# Multi-Omics Characterization of a Human Stem Cell-Based Model of Cardiac Hypertrophy

**DOI:** 10.3390/life12020293

**Published:** 2022-02-16

**Authors:** Markus Johansson, Benjamin Ulfenborg, Christian X. Andersson, Sepideh Heydarkhan-Hagvall, Anders Jeppsson, Peter Sartipy, Jane Synnergren

**Affiliations:** 1Systems Biology Research Center, School of Bioscience, University of Skövde, SE-541 28 Skövde, Sweden; benjamin.ulfenborg@his.se (B.U.); sepideh.hagvall@astrazeneca.com (S.H.-H.); peter.sartipy@astrazeneca.com (P.S.); jane.synnergren@his.se (J.S.); 2Department of Molecular and Clinical Medicine, Institute of Medicine, The Sahlgrenska Academy at University of Gothenburg, SE-413 45 Gothenburg, Sweden; anders.jeppsson@vgregion.se; 3BioPharmaceuticals R&D Cell Therapy, Research and Early Development, Cardiovascular, Renal and Metabolism (CVRM), BioPharmaceuticals R&D, AstraZeneca, SE-413 83 Gothenburg, Sweden; christian.andersson1@astrazeneca.com; 4Bioscience, Research and Early Development, Cardiovascular, Renal and Metabolism (CVRM), BioPharmaceuticals R&D, AstraZeneca, SE-413 83 Gothenburg, Sweden; 5Department of Cardiothoracic Surgery, Sahlgrenska University Hospital, SE-413 45 Gothenburg, Sweden

**Keywords:** cardiac hypertrophy, cardiomyocytes, disease model, endothelin-1, stem cells, transcriptomics, proteomics

## Abstract

Cardiac hypertrophy is an important and independent risk factor for the development of cardiac myopathy that may lead to heart failure. The mechanisms underlying the development of cardiac hypertrophy are yet not well understood. To increase the knowledge about mechanisms and regulatory pathways involved in the progression of cardiac hypertrophy, we have developed a human induced pluripotent stem cell (hiPSC)-based in vitro model of cardiac hypertrophy and performed extensive characterization using a multi-omics approach. In a series of experiments, hiPSC-derived cardiomyocytes were stimulated with Endothelin-1 for 8, 24, 48, and 72 h, and their transcriptome and secreted proteome were analyzed. The transcriptomic data show many enriched canonical pathways related to cardiac hypertrophy already at the earliest time point, e.g., cardiac hypertrophy signaling. An integrated transcriptome–secretome analysis enabled the identification of multimodal biomarkers that may prove highly relevant for monitoring early cardiac hypertrophy progression. Taken together, the results from this study demonstrate that our in vitro model displays a hypertrophic response on both transcriptomic- and secreted-proteomic levels. The results also shed novel insights into the underlying mechanisms of cardiac hypertrophy, and novel putative early cardiac hypertrophy biomarkers have been identified that warrant further investigation to assess their potential clinical relevance.

## 1. Introduction

Cardiovascular diseases (CVD) are estimated to be responsible for 31% of all deaths worldwide [1]. Cardiac hypertrophy is, in its pathological form, irreversible and is initially a process where the heart is compensating for an increased workload, commonly due to extrinsic pathological stimuli, such as myocardial infarction, aortic stenosis, chronic hypertension, or other conditions, that alter the homeostasis of the heart [2]. In order to increase the quality of life and outcomes for affected patients, new and improved treatment options are needed.

To investigate the changes that occur during the development of cardiac hypertrophy, in vitro models based on human pluripotent stem cells offer an attractive alternative. In the past decade, the availability of human-induced pluripotent stem cell-derived cardiomyocytes (hiPSC-CMs) with high similarity to bona fide CMs has increased rapidly [3,4,5,6]. Several strategies to induce hypertrophy in hiPSC-CMs have also been developed using neurohormonal stimulation- or stretch-induced methods [7,8,9]. Endothelin-1 (ET-1) is frequently used in neurohormonal models and has been shown to induce hypertrophy within 24 h [10,11,12]. ET-1 binds to G-protein coupled receptors, which downstream leads to the activation and production of inositol-1.4.5-triphosphate (IP_3_), resulting in the release of intracellular calcium (Ca^2+^). The release of intracellular Ca^2+^ has been shown to mediate hypertrophic signaling via the calcineurin-NFAT pathway and potentially through other signaling pathways [13,14]. ET-1 stimulation is also an inducer of MAPK signaling, which is involved in CM cell growth [15]. Although these in vitro models have provided important knowledge about mechanistic aspects of cardiac hypertrophy, additional characterization is needed to support their further use as relevant disease models [16].

Two well-known protein markers altered in cardiac hypertrophy are ANP and BNP. These proteins are natriuretic peptides that are secreted from CMs in response to the stretching of the myocardium [17]. They both lower vascular tone and cardiac output as well as regulate blood volume through inhibition of aldosterone synthesis and renin secretion [18]. There are also specific signaling pathways that are involved in cell growth, cytoskeleton remodeling, fibrosis, and metabolism that are altered during the development of cardiac hypertrophy [19,20,21,22,23]. With translatable in vitro models, we can expand our knowledge of the molecular regulation of cardiac hypertrophy and identify novel multimodal biomarkers that can aid in the early detection of the disease.

RNA sequencing (RNA-seq) analysis has become less expensive and more standardized over the past decade and is ideal for studying the whole transcriptome and for the investigation of individual genes as well as groups of genes. It provides a snapshot of all transcribed genes in a sample and can reveal underlying mechanisms associated with cellular phenotype or function.

Recent advances in proteomic technologies have facilitated the identification of biomarkers in the secretome using high-throughput affinity proteomics, which can be used for the detection of proteins in serum, plasma, and conditioned cell culture media [24]. Using this method, it is possible to quantify the proteins that are secreted by the CMs and subsequently identify their possible associations with cardiac hypertrophy. Moreover, the identification of novel multimodal biomarkers that can detect early signs of cardiac hypertrophy would also be more approachable. A major advantage of this technique over mass spectrometry is that it requires very low sample volume, and it can be applied using fluids with a high content of albumin, such as cell culture media, which has been a limitation for other methods [25].

In this study, we used hiPSC-CMs stimulated with ET-1 as a model system to explore changes in gene- and secreted protein expression. We used RNA-seq and high-throughput affinity proteomics together with ELISA to characterize the hypertrophic response of ET-1 stimulation in the cells. Moreover, Ingenuity Pathway Analysis (IPA) was applied to analyze altered canonical pathways, cardiac effects, and predicted upstream regulators based on the differentially expressed genes (DEGs).

## 2. Materials and Methods

### 2.1. Cardiomyocytes

Human CMs derived from the hiPSC line Cellartis^®^ ChiPSC22 were obtained from Takara Bio (Takara Bio Europe AB, Gothenburg, Sweden). The CMs were cryopreserved at day 19 following the onset of differentiation using the STEM-CELLBANKER^®^ (cat 11890, Amsbio, Cambridge, MA, USA). CMs were aliquoted into vials (6 million/vial) in the freezing procedure and then stored in a −150 °C freezer. Before freezing the CMs, they showed regular beating characteristics with more than 95 percent of cardiac troponin T-positive cells.

### 2.2. Flow Cytometry

Flow cytometry was performed according to the protocol previously described in Johansson et al. [10].

### 2.3. Hypertrophy Induction

There are several strategies to induce hypertrophy in CMs with ET-1. In this study, we have used the method previously described in [10]. Cryopreserved CMs differentiated by Takara Bio (Takara Bio Europe AB, Gothenburg, Sweden) were thawed and plated into 8 wells in a 24-well plate. The thawing media consisted of Advanced RPMI (cat 12633012, ThermoFisher Scientific, Waltham, MA, USA), GlutamMAX™ (cat 35050061, ThermoFisher Scientific), B-27™ (cat 17504044, ThermoFisher Scientific), and Y27632 (cat Y0503, Sigma–Aldrich, St. Louis, MO, USA). Twenty-four hours after thawing and plating the cells, media was changed (0.8 mL/well). The culture media used was the same as the thawing media except that Y27632 was excluded. To let the CMs recover from the thawing procedure, they were cultured for 6 days, with the medium change every other day, before starting the hypertrophy stimulation with ET-1 (10 nM). At the start of stimulation, the CM showed normal beating characteristics and morphology. For the hypertrophy stimulation, the media was changed to the standard CM culture media with the addition of ET-1 (10 nM). After 8 h, CMs from one well were collected. At 24 h, CMs from a second well were collected. In the remaining two wells (the 48- and 72-h timepoint wells), the media was changed to new CM culture media with ET-1. At 48 h, CMs from the 48 h timepoint well were collected, and the media was changed in the 72-h timepoint well. The exact same procedure was performed for the control cells but with ET-1 excluded from the media. The whole procedure was repeated at five separate timepoints (*n* = 5).

### 2.4. RNA-Seq Analysis

Library construction was performed using Illumina Truseq stranded total RNA with the Illumina Ribozero method. Clustering was done by ‘cBot’, and samples were sequenced on a NovaSeq6000 (NovaSeq Control Software 1.6.0/RTA v3.4.4) with a 2 × 51 setup using ‘NovaSeqXp’ workflow in the ‘S1′ mode flowcell. The Bcl to FastQ conversion was performed using bcl2fastq_v2.19.1.403 from the CASAVA software suite. The quality scale used was Sanger/phred33/Illumina 1.8+. Processing of FASTQ files was carried out by the SciLifeLab National Genomics Infrastructure at the Uppsala Multidisciplinary Center for Advanced Computational Science, Sweden. Sequenced reads were quality controlled with the FastQC software and pre-processed with Trim Galore. Processed reads were then aligned to the reference genome of Homo sapiens (build GRCh37) with the STAR aligner. Read counts for genes were generated using the featureCounts library and normalized FPKM values calculated with StringTie. Technical documentation on the RNA-seq pipeline can be accessed here: https://github.com/SciLifeLab/NGI-RNAseq (accessed at 13 January 2022). Raw and processed data are available for download at ArrayExpress (https://www.ebi.ac.uk/arrayexpress/, accessed at 13 January 2022) accession number: E-MTAB-11030

### 2.5. Differential Expression Analysis

The raw gene count data, including 63,677 transcripts from 40 samples, were imported into R [26] for further analysis, and statistical testing for differential expression was carried out with the quasi-likelihood F-test in the edgeR package [27]. Only genes with >1.5-fold change (FC) were included in the results. A false discovery rate (FDR) of ≤0.05 was considered statistically significant.

To explore the overlap of DEGs between the different time points, Venn diagrams of up- and downregulated genes, respectively, were generated using InteractivVenn [28].

Cluster analysis was performed using the kmeans R package. Only genes annotated with the ‘cardiac muscle hypertrophy’ gene ontology term (GO:0003300) were selected and included in the clustering analysis. The input data for the clustering was mean FPKM values for ET-1 stimulated samples. To determine the number of clusters to be used in k-means, the gap statistic method, which considers the within-cluster dispersion, was applied. In total, 11 clusters were suggested as the optimal for the number of clusters (Appendix A).

### 2.6. Upstream Regulators

For in-depth analysis of DEGs, we used the QIAGEN’s Ingenuity^®^ Pathway Analysis (IPA^®^, QIAGEN Redwood City, www.qiagen.com/ingenuity, accessed at 13 January 2022) software for upstream regulatory analysis of the identified DEGs.

All DEGs with an absolute FC > 1.5 were used as input in the analysis. The Z-score is a measurement to predict the activation or inhibition of regulators based on the relationships with the dataset genes and direction of change in dataset genes (positive values: activated; negative values: inhibited). A Z-score > 2 is considered statistically significant [29]. The overlap *p*-value measures whether there is a statistically significant overlap between the DEGs and the genes that are regulated by an upstream regulator. It was calculated using Fisher’s exact test, and significance attributed to *p*-values < 0.01. We limited the upstream regulator analysis to only include upstream regulators that were genes, RNAs, or proteins.

### 2.7. Transcription Regulators

In IPA, filtering was applied to only include DEGs that are TRs. Of this subset of DEGs, genes with evidence of a direct association with hypertrophy of the heart were selected for the analysis.

### 2.8. Canonical Pathway and Cardiac Clinical Outcome Analysis

For the canonical pathway analysis, all DEGs with FC above 1.5 or below −1.5 were selected as input in the analysis. A canonical pathway had to be significantly activated or inactivated at least in one of the 4 time points to be included in the resulting output. For the cardiac clinical outcome analysis, the IPA tox function was used. Clinical outcomes with *p* < 0.05 were considered as enriched. The activation Z-score was used to predict the activation state of the clinical outcome, and Z-score > 2 was considered statistically significant.

### 2.9. Affinity Proteomics and ELISA

Conditioned media from all the samples was sent to Olink (Uppsala, Sweden) for PEA analysis. In short, the PEA involves binding distinct polyclonal oligonucleotide-labeled antibodies to the target protein, followed by quantification by real-time quantitative PCR. A total of 40 undiluted samples were analyzed on 3 different Olink panels (Cardiovascular II, Cardiovascular III, and the Inflammation panel).

The protein measurement results from the PEA analysis are reported as normalized protein expression, NPX, an arbitrary unit in the log_2_ scale. NPX data were imported and analyzed in R. Missing values were imputed by the mean of remaining replicates for the sample group. Protein measurements were background corrected by first de-logging the NPX values and subtracting the negative control sample for all samples. The data were then log_2_-transformed again. Values below 0 after background-correct were replaced by the protein-specific limit of detection (LOD) value minus one (when all values were below 0), or the lowest NPX value for the protein minus one (when log_2_ values were present). The data were then filtered to remove proteins where the expression was below LOD in all biological samples, after which 115 proteins remained. Proteins were retained if all replicates were in at least one sample group which had NPX values above LOD. Statistical analysis of differential expression was performed with limma, and *p*-values were corrected for multiple testing with the Benjamin–Hochberg method. Proteins with an adjusted *p*-value < 0.05 and absolute fold change > 1.5 were considered significant.

Due to high concentration, a few proteins could not be analyzed with the affinity proteomics method, and these were analyzed separately with ELISA kits, according to the manufacturer’s instructions (Table 1).

### 2.10. Correlation Analysis

All the gene–protein pairs that had the protein significantly differentially expressed at one time point or more were included in the correlation analysis. Two different correlation analyses were performed. One correlation was calculated between mRNA levels and protein levels from ET-stimulated cultures at the same time point. The second correlation analysis was performed comparing mRNA levels, and proteins levels from ET-stimulated cultures with the protein levels shifted one time point. The time-shifted analysis resulted in the following comparisons: mRNA 8 h vs. protein 24 h, mRNA 24 h vs. protein 48 h, and mRNA 48 h vs. protein 72 h. Analyses were performed using Pearson’s correlation coefficients (GraphPad Prism 9.2.0. GraphPad Software, San Diego, CA, USA).

## 3. Results

### 3.1. Cardiomyocyte Homogeneity

The differentiated hiPSC-CMs at day 19 showed a high percentage (97%) of cTnT+ cells when analyzed with flow cytometry (Appendix A).

### 3.2. Transcriptomic Perturbations in Response to ET-1 Stimulation

Control and ET-1 stimulated culture samples were collected at 8, 24, 48, and 72 h. The experiment was repeated 5 times, resulting in 40 samples in total. All samples were analyzed with RNA-seq, and differential expression analysis was performed. The largest number of differentially expressed genes (DEGs) was identified at the 8-h time point. The number of significantly upregulated DEGs (FC > 1.5) were 1239, 536, 663, and 467 at time points 8, 24, 48, 72 h (h), respectively (Figure 1A). For significantly downregulated DEGs (FC > 1.5), the corresponding numbers were 1543, 845, 787, and 261 at 8, 24, 48, and 72 h, respectively (Figure 1A). The same trend with the highest number of DEGs at the beginning of the ET-1 stimulation was observed when setting the cutoff to FC > 2.0 (Figure 1A). The Venn diagrams visualize the number of overlapping DEGs between the time points (Figure 1B). For upregulated DEGs, 32 genes were overlapping between all time points, and for downregulated genes, the corresponding number was 27. The overlapping up- and downregulated genes are presented in Table 2 and Table 3.

All the overlapping genes from the Venn diagrams were analyzed using IPA to investigate if they have a known connection to hypertrophy. Of the 59 genes that were overlapping between all time points, four were included in the ’Enlargement of heart’ classification from IPA. These were *SLC6A4*, *ADRB3*, *CTSV*, and *PPP1R13L* (Figure 1D).

In Figure 1C, the top 20 up- and downregulated DEGs at each time point are presented, and a full list of DEGs is shown in Appendix A. Non-protein coding genes were excluded from these gene lists. Many genes of particular interest from a cardiac hypertrophy perspective are represented in these lists. At the 8 h time point, *APELA*, *ADRB3*, *MAP2K6* were among the most downregulated genes. *APELA*, a gene that codes for a peptide hormone that binds to the apelin receptor, showed the largest downregulation after ET-1 stimulation with a log_2_ FC-values above six (*p* < 9.3 × 10^−6^). Other genes associated with hypertrophy and that were among the top 10 most upregulated genes were *PDYN*, *PNOC*, and *TGFA*. At 24 h, the log_2_ FC for *PDYN* was 4.8 (*p* = 4.27 × 10^−192^), which is the highest of all genes at that time point. *PNOC* was significantly upregulated at all time and was among the top 10 most upregulated at 24, 48, and 72 h, with the largest log_2_ FC of 3.6 at 24 h (*p* = 7.2 × 10^−79^). *TGFA* was also significantly upregulated at all time points and among the top 10 most upregulated at 24 and 48 h. The log_2_ FC at 48 h was 2.0 (*p* = 3.7 × 10^−63^).

### 3.3. Pathway Analysis

To explore enriched canonical pathways among the DEGs and assess changes over time, all DEGs (FC > 1.5) were analyzed using IPA. After filtering out pathways with an absolute Z-score > 2, 78 pathways were significantly activated or inhibited at one or more of the four time points (Figure 2A). At the 8-h time point, the highest number of activated pathways (27 pathways with Z-score > 2) was observed, and at the 24-h time point, the highest number of inhibited pathways (15 pathways with Z-score < −2) was identified. One pathway of high relevance for our study was the ‘Cardiac hypertrophy pathway’ (enhanced). This pathway, like many other pathways, was activated after 8 h and subsequently downregulated at the later time points (Figure 2A,B). At 8 h, a total of 92 genes were differentially expressed in this pathway, and at the following time points (24, 48, and 72 h), these numbers were 28, 34, and 20, respectively.

Cytoskeletal changes are integral parts of cardiac remodeling, a hallmark of cardiac hypertrophy, and ‘Actin cytoskeletal signaling’ was the most enriched canonical pathway at 8 h (Z-score = 4.6, *p* = 1.4 × 10^−4^). Similar to ‘Cardiac hypertrophy signaling’, ‘Actin cytoskeleton signaling’ showed the highest number of DEGs after 8 h (in total 46), and 16, 17, and 9 DEGs at 24, 48, and 72 h, respectively (Figure 2A,C). Pathways that showed strong activation at later time points were the ‘Superpathway of Inositol Phosphate Compounds’ (highest Z-score at 72 h), ‘PPARα/RXRα Activation’, and ‘Apelin Cardiomyocyte Signaling Pathway’. ‘PPARα/RXRα’ was the canonical pathway with the largest difference in Z-score when comparing 8 to 72 h, ranging from −1.671 to 4.12 (Figure 2A).

### 3.4. Upstream Regulators

An upstream regulator analysis was performed using IPA and the DEGs as input to identify transcription factors, genes, or proteins that could explain the observed changes in gene expression. In Figure 3A, the top 10 upstream regulators, both activating and inhibiting, are presented. In total, IPA identified 275 upstream regulators (238 activated and 37 inhibited) at 8 h (Appendix A). Of these, 40 were differentially expressed in our dataset. Not surprisingly, ET-1 was one of the upstream regulators with the highest absolute prediction scores (Z-score = 4.9), which demonstrates that our cell-based in vitro model mimics the in vivo response to ET-1, based on IPA’s current knowledge base. At 24 h, in total, 89 upstream regulators were identified (65 activating and 24 as inhibiting). Two of the inhibiting regulators were also differentially expressed in our dataset, *E2F2* and *LMNB1*. At 48 h, 160 upstream regulators were identified (62 activating and 98 inhibiting) (Appendix A). Nine of these were also differentially expressed in our dataset. At 72 h, 10 upstream regulators were identified (6 activating and 4 inhibiting). *S100A8*, one of the inhibiting regulators, was also identified as differentially expressed (downregulated) in our dataset. There was no overlap of any upstream regulator between all the time points. However, some upstream regulators overlapped between two or more time points (Figure 3B and Appendix A).

### 3.5. Transcription Regulators

From the upstream regulator analysis, all differentially expressed transcription regulators (TRs) were selected for their association with the progression of the cardiac hypertrophy response. At 8 h, 15 TRs were differentially expressed, and of those, five were directly associated with ‘Hypertrophy of the heart’ (Figure 3C). Notably, in total, 180 of its target genes, *MYC*, were overlapping with the DEGs in our dataset (Figure 3C). A GO enrichment analysis of target genes of *MYC* showed ‘Regulation of apoptotic process’ as the most enriched GO term (*p* = 3.25 × 10^−13^). For *SRF*, the most enriched term was ‘Muscle contraction’ (*p* = 6.25 × 10^−8^). *TBX5* showed the most heart-specific enriched GO terms. Among the top enriched GO terms were ‘Cardiac muscle tissue development’ (*p* = 5.98 × 10^−6^), ‘Heart morphogenesis’ (*p* = 1.63 × 10^−5^) and ‘Response to muscle stretch’ (*p* = 3.41 × 10^−5^). At 24 h, two TRs were identified as differentially expressed, and one of them (*TBX5*) was associated with ‘Hypertrophy of cardiac-like myocytes’ (Figure 3D). At 48 h, 4 TRs were differentially expressed. One of them, *FOXM1*, is associated with ‘Hypertrophy of left ventricle’ (Figure 3E). Many of the enriched GO terms among the target genes of *FOXM1* were involved in the regulation of the cell cycle. At the last time point, 72 h, no TRs were identified as differentially expressed.

### 3.6. Cardiac Clinical Outcomes

To link our experimental data to clinical pathology endpoints, IPA’s tox function tool was applied, and the overlap between DEGs in our data and sets of genes associated with various clinical pathology endpoints was investigated. At 8 h, 71 endpoints had a significant overlap with DEGs in our data (Figure 4A). The endpoint with the highest gene overlap was ‘Enlargement of heart’ (*p* = 1.69 × 10^−13^). Four of the significant endpoints were also identified as activated (Z-score > 2). These were ‘Congenital heart disease’, ‘Atrial or ventricular septal defect’, ‘Ventricular septal defect’, and ‘Atrial septal defect’. At 24 h, 81 cardiac endpoints had a significant gene overlap with our set of DEGs, and the most significant gene overlap was also associated with ‘Enlargement of heart’ (*p* = 3.87 × 10^−9^) (Figure 4B). At 48 h, there were fewer significant cardiac endpoints identified, and, in total, 18 were shown as significant, and among these, ‘Hypertrophy of heart’ showed a notably low overlap *p*-value (*p* = 0.0005) (Figure 4C). At the last time point (72 h), only the endpoint named ‘Damage of cardiac muscle’ was significant (*p* = 0.036) (Figure 4D).

### 3.7. Cluster Analysis

Cluster analysis using k-means was performed using a set of hypertrophy-associated genes (GO term 0003300, ‘Cardiac muscle hypertrophy’) to identify genes with correlated expression profiles during ET-1 stimulation. Using the k-means clustering algorithm, 11 distinct clusters were identified (Appendix A. The genes in cluster 2 showed a downregulation between time point 8 h compared with the later time points (Figure 5A). The majority of the genes in this cluster are involved in the positive regulation of cardiac hypertrophy. The two genes in cluster 4, *AKAP1* and *FBX032*, which are negative regulators of cardiac hypertrophy, had their peak expression at 24 h and were decreased at the later time points (Figure 5B). Cluster 5 was of particular interest for this analysis, and the genes in this cluster showed relatively stable expression levels when comparing the different time points (Figure 5C). These genes are involved in both positive and negative regulation of cardiac hypertrophy. Genes in cluster 7 showed increased expression over time, and the child GO terms associated with these genes show conflicting regulation of cardiac hypertrophy; some are positive regulators, whereas others are negative regulators (Figure 5D). The genes in cluster 9 showed the highest expression values at the 24 h time point, and except for *EDN1*, they sustained a higher expression compared to the 8 h time point (Figure 5E). The cluster analysis showed that hypertrophy-associated genes have various expression patterns, which also change over time.

### 3.8. Protein Expression

The proteomics analysis of the conditioned cell media revealed several differentially expressed secreted proteins. In total, 23 proteins were differentially expressed (FC > 1.5, *p* < 0.05) in the ET-1 stimulated cells at some time point. Eighteen were identified from the affinity proteomics analysis and five from the ELISA analysis (Figure 6A,C). The number of differentially expressed proteins at 8, 24, 48, and 72 h were 5, 12, 11, and 9, respectively (Table 4). The lowest number of differentially expressed proteins were observed at 8 h, which is an opposite pattern versus the gene expression results that showed the highest number of DEGs at 8 h. Interestingly, only 2 of the 23 differentially expressed proteins were downregulated. These proteins were Interstitial collagenase (*MMP1*) and Collagen type 1 alpha (*COL1A1*) (Figure 6A).

At 8 h, ‘CCL-4’ was the most upregulated protein with a FC of 4.8 (*p* = 0.006). This chemokine was among the highest upregulated proteins at 24 h as well (FC = 7.4, *p* = 0.003), together with ‘CCL-3’ (FC = 7.7, *p* = 0.002) (Figure 6A). Another protein that was significantly upregulated at both 8 and 24 h was ‘Proheparin-binding EGF-like growth factor’, which showed a more than 2-fold upregulation at both time points (*p* < 0.001) (Figure 6A). At 48 h, ‘Chitinase-3-like protein 1’ was the most upregulated protein with a FC of 3.2 (*p* = 0.0009), and it was also upregulated at 24 h (FC = 2.1, *p* = 0.02). At 72 h, several additional proteins were detected in the analysis. The top 2 most upregulated proteins at 72 h were ‘Heme oxygenase 1’ and ‘Thrombomodulin’, which had a FC of 2.8 (*p* = 0.03) and 2.6 (*p* = 0.04), respectively. (Figure 6A). 

Using ELISA, we identified additional proteins that due to high concentrations could not be quantified by the affinity proteomics method. These were ANP, ProBNP, HSPG2, PAI1, and COL1A1. The concentration of ANP was significantly increased in the stimulated groups at 24 (FC = 1.5, *p* < 0.01) and 48 h (FC = 1.7, *p* < 0.01). ProBNP concentration was significantly increased at 8 (FC = 4.0, *p* = 0.02), 24 (FC = 2.6, *p* < 0.01), and 48 h (FC = 2.4, *p* < 0.01). HSPG concentration was significantly increased at 72 h (FC = 1.8, *p* = 0.05). PAI1 was significantly increased with a FC > 1.5 at 8, 24, and 48 h; the highest FC was observed at 24 h (FC = 9.3, *p* < 0.01). COL1A1 concentration was significantly decreased at 48 h (FC = −1.6, *p* = 0.02) (Figure 6C).

Out of the 23 proteins that were identified as differentially expressed, 10 were annotated with ’Enlargement of heart’, according to the IPA knowledge database. These proteins were COL1A1, HBEGF, HMOX1, HSPG2, MMP1, NPPA, PGF, SERPINE1, TNFRSF11B, and VEGFA (Figure 6D).

A correlation analysis was performed comparing the gene expression and secreted protein expression in ET-1 stimulated cultures. To explore if there was a time delay from an mRNA molecule to a produced and secreted protein that can be measured, we performed two different correlation analyses—one with mRNA expression and protein expression from the same time point and one with the protein expression shifted one time point (e.g., mRNA at 8 h vs. protein at 24 h). Interestingly, the time-shifted analysis showed a strong correlation (r > 0.8) for 12 of the 23 analyzed gene–protein pairs. For the “no time-shift” analysis, the corresponding number was five (Table 5).

## 4. Discussion

In this study, we have used a cell-based model of cardiac hypertrophy by stimulating hiPSC-derived CMs with ET-1 and characterized the response on transcriptome and secretome levels. The results were in line with previous studies and provided further insights into the molecular machinery underlying the initiation and development of cardiac hypertrophy [10,11,30]. Our results illustrated that known cardiac hypertrophy genes were differentially expressed in our model and signaling pathways that are important in a hypertrophic response were enriched in the ET-1 stimulated cultures. Our analysis of secreted proteins also showed that there were several key proteins associated with cardiac hypertrophy that were differentially expressed in our model, and potential new candidate biomarkers or putative drug targets were identified.

hiPSC-derived CMs have been used for the development of several in vitro disease models [31,32,33]. They have shown to be promising for studies of cardiac hypertrophy, with characteristics resembling the in vivo situation. For example, we and others have previously reported that the hiPSC-derived CMs increase in size, have altered metabolism, and a transcriptional pattern indicative of hypertrophy when exposed to substances such as ET-1 and phenylephrine [10,11,12]. Although information is still lacking regarding the optimal maturation stage of the CMs for use in a hypertrophy model, our results demonstrated that cells cultured for at least 20 days produce a robust hypertrophic response [10].

Our analysis identified an increased expression of many genes that are related to cardiac hypertrophy already at 8 h of ET-1 stimulation. In fact, the largest numbers of DEGs were observed at the earliest time point (8 h) and then decreased over the 24, 48, and 72 h time points. This change was observed when analyzing both DEGs with a FC of 2 and 1.5. The trend in the data with a lower number of DEGs over time was in contrast to the stretch model of cardiac hypertrophy, where an increase in DEGs is observed with extended treatment [34]. Notably, our gene expression data at the 8 h time point were most similar to the latest time point (48 h) in the stretch model. It is possible that the neurohormonal approach that we used resulted in a more rapid hypertrophic response compared to the stretch model. It should be noted, however, that there is also a difference between the cellular source and species used in the studies (neonatal rat CM vs. human hiPSC-derived CMs), adding further complexity when interpreting data across the studies.

Some of the genes with the highest fold change that were picked up from the differential expression analysis were *PDYN*, *PNOC*, and *TGF*. The *PDYN* gene was upregulated at all time points and was also the most upregulated gene at 24 h and the second most upregulated at 48 h and 72 h. The *PDYN* gene encodes the protein Proenkephalin-B, which is a preproprotein that is processed into several opioid peptides. They bind primarily to the K-opioid receptors that have been found to be widely expressed in the human heart [35,36]. Information about endogenous opioid peptides and their role in the cardiac hypertrophy development is sparse. However, there are data that support the hypothesis that the opioid system initially has a cardioprotective role. For example, in rat CMs, it has been shown that the use of a K-opioid receptor agonist protects against hypertrophy and fibrosis when stimulating the cells with the prohypertrophic agent isoprenaline [37]. Taken together with the data from the present study, *PDYN* appears to be a key candidate for further research in order to understand its potential involvement in the development of cardiac hypertrophy.

The gene *PNOC*, which was also among the top DEGs at all time points, codes for the protein prepronociceptin. Studies using other in vitro models of cardiac hypertrophy have reported upregulation of *PNOC* when neonatal rat CMs were stimulated with isoprenaline and phenylephrine. However, stimulation with ET-1 did not upregulate *PNOC* [38]. In contrast, in our model, using hiPSC-derived CMs, we saw a significant upregulation of this gene after ET-1 stimulation. The effect of the upregulation of *PNOC* in humans is not well studied, and in pre-clinical models, the data are inconsistent. For example, in rodents, *PNOC* has been shown to have inhibitory effects on cardiovascular parameters, while in sheep, it increases blood pressure [39,40]. More research is needed to elucidate if the distinct upregulation of *PNOC* forms part of any cardioprotective mechanisms or if it may play a role in the long-term progression of cardiac hypertrophy.

The most downregulated gene at all time points was *APELA*, encoding for an apelin receptor ligand. The apelin signaling in cardiac hypertrophy has been studied but whether apelin is beneficial or aggravating in this setting remains unclear [41,42,43,44]. In rats, administration of angiotensin II downregulates the expression of apelin within 24 h [45]. In humans, there are data suggesting a strong negative correlation between serum apelin levels and left ventricular hypertrophy [46].

The pathway analysis using IPA identified the largest number of significantly enriched pathways at the 8 h time point; the number then decreased over time. This could be a consequence of the number of DEGs since fewer DEGs make the analysis less sensitive. These observations are in line with our previous study [10]. ‘The Cardiac hypertrophy’ (enhanced) pathway was significantly enriched at 8 h, which was expected. However, already at 24 h, this pathway became inhibited, indicating that many of the genes in the pathway that were upregulated at 8 h were downregulated at 24 h. This switch in gene expression suggests a fast adaption or feedback response on a transcriptional level. Inhibition of hypertrophy pathways has been shown before when using in vitro models and hypertrophic stimulation for more than 24 h [7]. While many pathways were only enriched at the first time point in our data, we know from our previous study that the hypertrophy phenotype is persistent over time [10].

Our results showed similarities with previous studies. Signaling pathways that were enriched in our data have been previously reported by others, including actin cytoskeleton, angiopoietin, *IL-6*, *IL-8*, *HMGB1*, and metabolic pathways [7,34]. Interestingly, adrenergic signaling in CMs was downregulated in both our neurohormonal model as well as the stretch model described by Ovchinnikova et al. [7]. In heart failure, β-adrenergic signaling is downregulated, possibly as a compensatory mechanism. Today, inhibiting β-adrenergic signaling using β-antagonists is a cornerstone in the pharmacological treatment of heart failure [47]. We also found that *IL-6*, Oncostatin M, and JAK/STAT signaling were enriched at 8 h. An increase in the signaling of these pathways suggests that the STAT3 pathway is enriched, something that we also found in our analysis. Enrichment of the STAT3 pathway has been shown to promote cardiac hypertrophy by inducing the growth of CMs. Overexpression studies have also found that it increases the expression of hypertrophy genes, such as *NPPA*, *MYH7*, and *CTF1* [48].

From the upstream regulator analysis using IPA, several TRs that play a role in cardiac hypertrophy were identified as differentially expressed in our data set. These regulators affect, in the majority of cases, many genes that were differentially expressed in our data and may therefore represent interesting targets in drug discovery. *MYC* was one of the most active TRs that was differentially expressed in our study and is known to be important in early development, where it controls the proliferation of CMs [49]. In addition, it is upregulated when CMs are exposed to hypertrophic stimuli, and it can also induce hypertrophy by itself [50,51,52]. Neonatal rat CMs cultured in vitro and stimulated with ET-1 showed a significant hypertrophic response with the upregulation of *MYC*, a response attenuated when *MYC* was inhibited [53]. Another TR associated with cardiac hypertrophy that was upregulated in our stimulated CMs is *SRF*. *SRF* is known to be required for the induction of the “fetal gene program”, which is a set of genes that is characteristic of cardiac hypertrophy [54]. Overexpression studies in animals show that it regulates the fetal genes long before an increase in heart weight is observed [55]. We observed differential expression of some, but not all, of the genes in the fetal gene program (e.g., *NPPA*, *NPPB*, and *ACTA1*). We also analyzed some upstream regulators that are transcription factors. *TBX5*, which was differentially expressed at 8 and 24 h, regulates the expression of several genes that are specific to the heart [56]. Although *TBX5* was downregulated at 8 and 24 h, it is possible that this gene was upregulated at an even earlier time point. Importantly, GO enrichment analysis of the DEGs that *TBX5* regulates in our data set identified many GO terms associated with the heart. This suggests that *TBX5* may be a target for drugs against cardiac hypertrophy since it will be more specific to the heart.

As described in the differential expression analysis, we identified many genes that are of importance in cardiac hypertrophy, showing that our model resembles the in vivo situation in many aspects. Therefore, we went on to use the model for the identification of differentially expressed proteins that are secreted from the CMs, and that can be measured in the conditioned media.

We performed a focused proteomics analysis using Olinks CVD II, III, and Inflammation panels to identify differentially expressed secreted proteins. To our knowledge, this is the first time where secretome analysis has been performed on conditioned media from a human stem cell-based model of cardiac hypertrophy. The secretome studies of CMs reported to date have been performed using animal models or during differentiation of stem cells to CMs [57,58,59]. Twenty-three proteins were significantly differentially expressed at least at one of the time points in our study. Notably, 10 of them had the corresponding gene significantly differentially expressed also at some time point in the experiment. These genes/proteins were *CXCL8*, *HBEGF*, *LDLR*, *NPPA*, *NPPB RARRES2*, *SERPINE1*, *TNFRS11B*, *PGF*, and *PRSS8* (Figure 5). However, the potential time difference from mRNA to secreted protein should not be neglected and when the correlation analysis was performed with one time point shifted for the proteins (e.g., mRNA at 8 h vs. protein at 24 h), the observed correlation was substantially higher. In fact, when using the time shift approach, the majority of gene–protein pairs had a correlation value > 0.8. The rate of mRNA to protein is in part affected by the translation rate, the binding of regulatory elements, such as micro-RNAs, protein half-life modulation through the ubiquitin–proteasome pathway, and protein synthesis delay [60]. These are all factors that may explain the observed difference in the correlation between the different proteins in the study.

Seven secreted proteins were identified as differentially expressed at 72 h, and many of them showed a strong correlation between protein expression and gene expression. Chemerin, which is encoded by the gene *RARRES2*, was significantly upregulated at both gene and protein levels at 48 and 72 h and had an overall correlation between gene and protein expression for the ET-1 stimulated cells of r = 0.72. Chemerin has a well-known role in metabolic syndrome and may contribute to hypertension by acting on chemokine-like receptor 1 [61,62].

Two growth factors that are in the same family, vascular endothelial growth factor A and placental growth factor, were also significantly upregulated at 72 h. Both these proteins are associated with angiogenesis and vasculogenesis and may indicate that the heart compensates for the increase in work and size by trying to increase its vascular network [63,64]. This mechanism is observed in physiological cardiac hypertrophy, where the blood flow to the heart grows in proportion to the size of the heart. Our relatively pure CM population made it difficult to assess if the increase in these growth factors could stimulate the formation of new blood vessels. However, an increase could very well be an indicator of stress of the heart and possibly an early sign of hypoxia in the CMs [65].

In summary, we have shown that our model of cardiac hypertrophy based on hiPSC-CM and neurohormonal stimulation recapitulates important characteristics of cardiac hypertrophy on a transcriptional level. Pathway and cluster analyses also shed novel insight into the underlying mechanisms potentially regulating cardiac hypertrophy on a cellular level. Additionally, several secreted proteins were differentially expressed, and further investigations are warranted to determine their exact role in the hypertrophy response and how they can be used clinically in the future.

## 5. Conclusions

The extensive characterization of our in vitro model of cardiac hypertrophy showed a robust response on both transcriptome and secretome levels. The largest transcriptional response was identified already at the earliest assessed time point (8 h), while the number of identified DEGs decreased at later time points. The signaling pathway analysis showed enrichment of several pathways that play important roles in cardiac hypertrophy, e.g., actin cytoskeleton signaling and cardiac hypertrophy signaling. Interestingly, cardiac hypertrophy signaling was inhibited at the later time points, suggesting a fast adaptive response. Analysis of the secretome identified several differentially expressed proteins of high relevance for cardiac hypertrophy and, interestingly, for many of these proteins, a strong correlation was observed between the protein expression and the gene expression levels. The known hypertrophy markers ANP and proBNP were both significantly increased in the ET-1 stimulated cultures. The secretome analysis identified several proteins with potential as biomarker candidates for cardiac hypertrophy. However, more research is needed to conclude their specific role in the disease progression and how close the in vitro condition reflects the in-vivo counterpart.

## Figures and Tables

**Figure 1 life-12-00293-f001:**
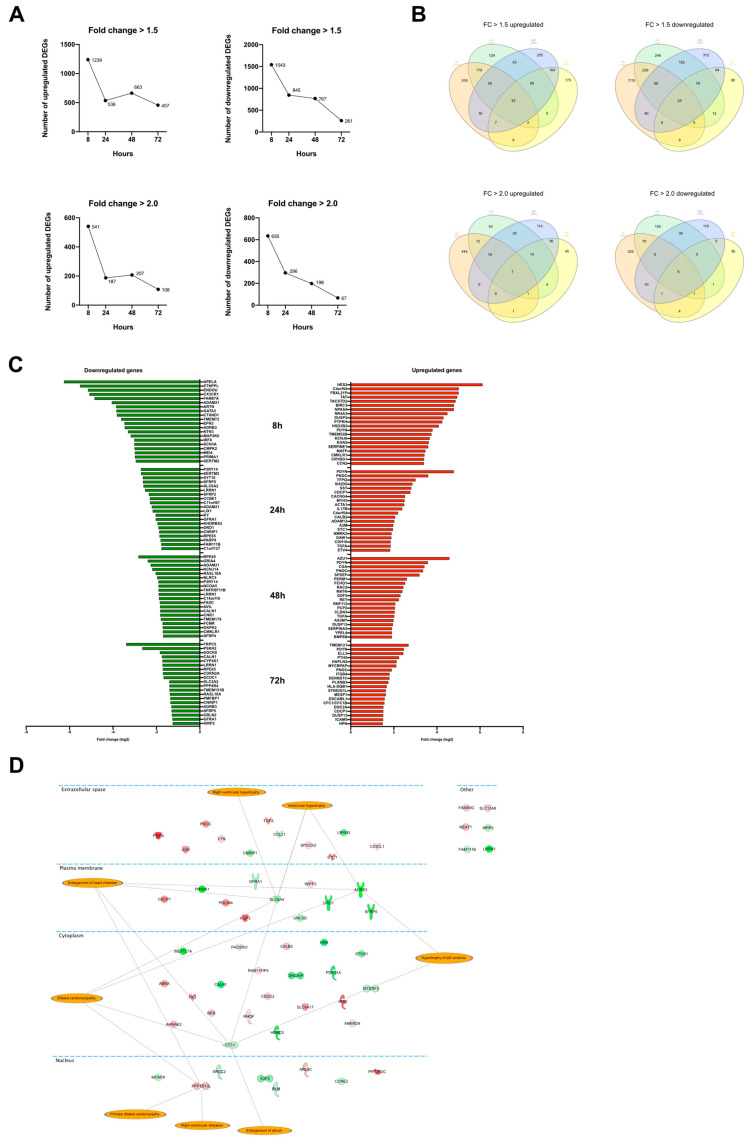
Differential expression analysis. (**A**) Graphs showing the number of DEGs at the different time points. Top row shows the number of DEGs with fold change >1.5. The two diagrams below show the number of DEGs with fold change >2.0. The *y*-axis shows the number of genes, and the *x*-axis, the time point. (**B**) Venn diagram illustrating the overlap of DEGs between the time points. (**C**) Bar plot showing the top 20 most up-and downregulated genes at 8, 24, 48, and 72 h. (**D**) Differentially expressed genes that were overlapping between all four time points. The orange highlights represent terms associated with the annotation ‘Enlargement of heart’ in the IPA’s knowledge database.

**Figure 2 life-12-00293-f002:**
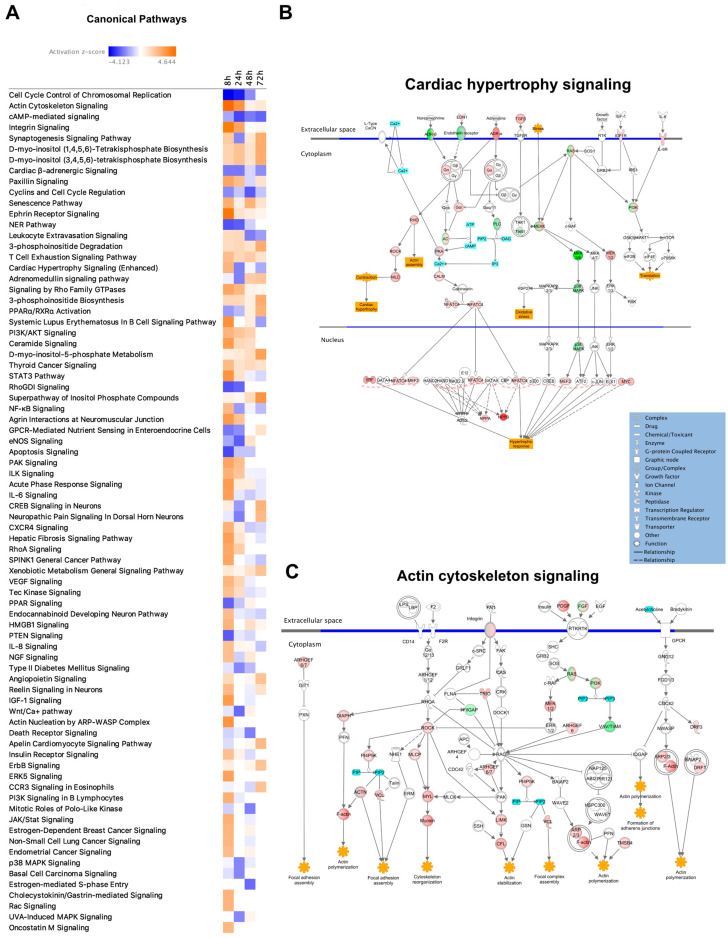
Canonical pathway analysis. (**A**) Table showing all the pathways that were significantly (*p* < 0.05) enriched or inhibited at some time point and with a Z-score > 2. Color scale ranges from blue to orange, with blue representing the strongest inhibition and dark orange representing the strongest enrichment. (**B**) Diagram of the ‘Cardiac hypertrophy signaling pathway’. Red and green represent up- and downregulated gene/gene groups, respectively. (**C**) Diagram of the ‘Actin cytoskeleton signaling pathway’.

**Figure 3 life-12-00293-f003:**
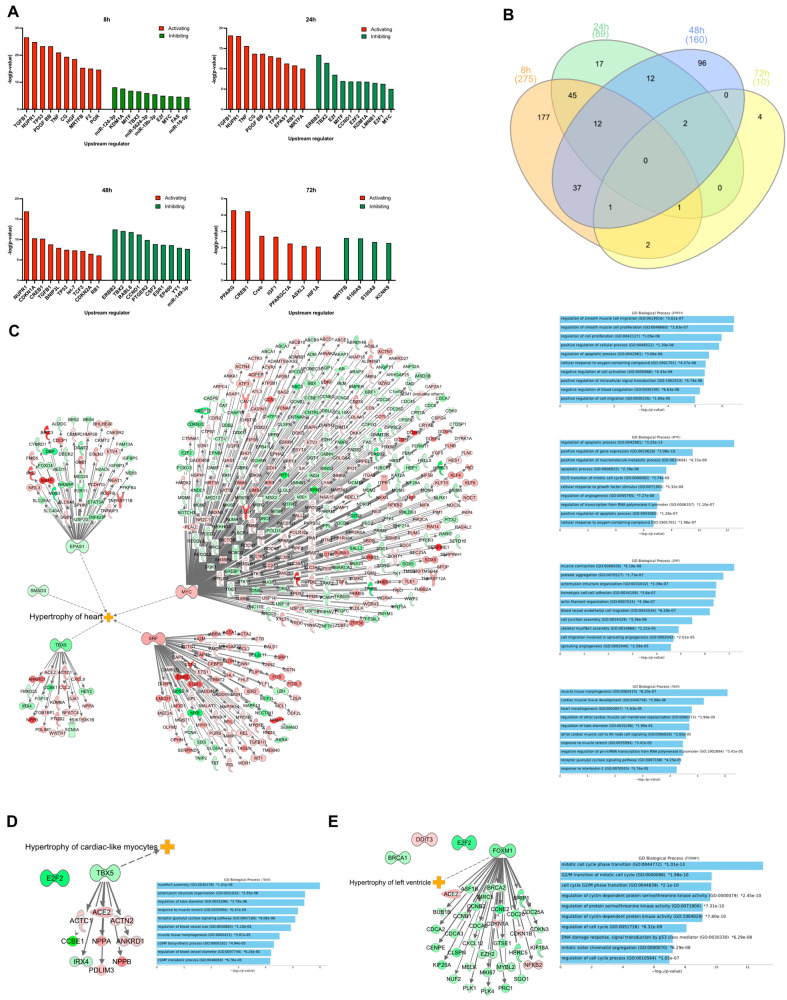
Upstream and transcription regulator analysis. (**A**) Graphs show the predicted top 10 most activated or inhibited upstream regulators at 8, 24, 48, and 72 h. The *y*-axis shows the −logP value of the overlap between the genes in the dataset and genes regulated by an upstream regulator. Red and green bars represent activated and inhibited regulators, respectively. (**B**) Venn diagram showing the overlap of upstream regulators between the different time points. (**C**) Transcription regulators that were differentially expressed at 8 h. Dotted line shows transcription regulators that have an association with IPAs hypertrophy of heart classification. (**D**) Transcription regulators that were differentially expressed at 24 h. Dotted line shows which transcription regulator that have an association with IPA’s ‘Hypertrophy of cardiac-like myocytes’ classification. (**E**) Transcription regulators that were differentially expressed at 48 h. Dotted line shows which transcription regulator that has an association with IPA’s ‘Hypertrophy of left ventricle’ classification. Tables to the right of the (**C**–**E**) show the GO biological processes terms that were enriched when analyzing the TRs target genes that were differentially expressed at the indicated time point.

**Figure 4 life-12-00293-f004:**
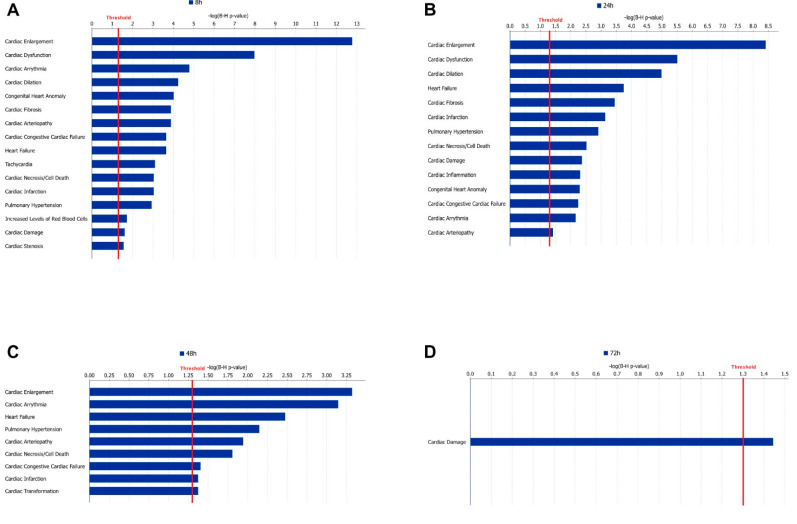
Cardiac clinical outcome analysis. Panels (**A**–**D**) show the significant (*p* < 0.05) cardiac clinical endpoints from IPAs cardiotoxic analysis at 8, 24, 48, and 72 h, respectively. The *x*-axis shows the −logP value of the overlap, and the red line represents the significance threshold.

**Figure 5 life-12-00293-f005:**
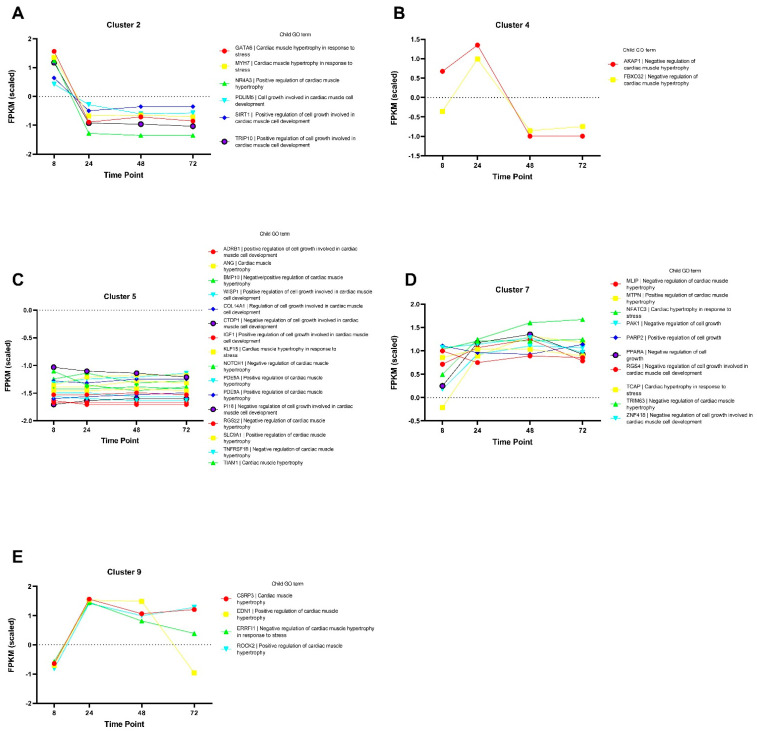
Cluster analysis of genes annotated with the GO term 0003300 ‘Cardiac muscle hypertrophy’. Panels (**A**–**E**) show the genes included in a particular cluster generated with the k-means cluster algorithm. The *y*-axis shows normalized FPKM values, and the *x*-axis shows the different time points. The tables to the right of the graphs show a description of the child GO term associated with each gene.

**Figure 6 life-12-00293-f006:**
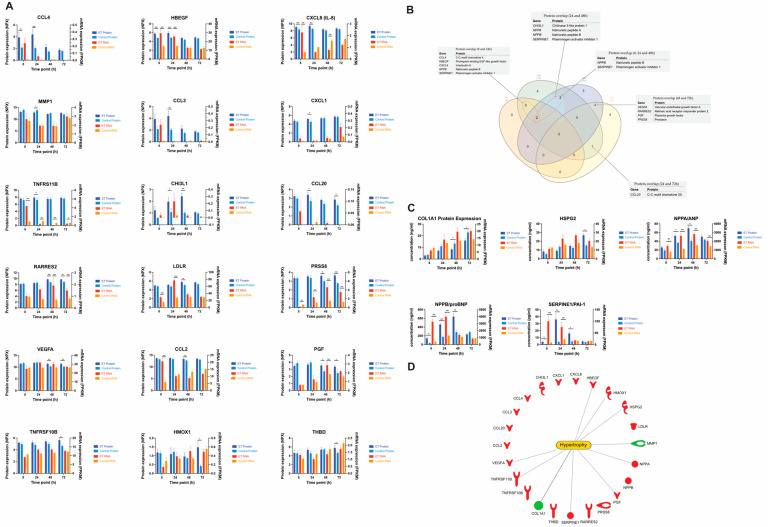
Proteomics. (**A**) All differentially expressed proteins (FC > 1.5 *p* < 0.05) from the affinity proteomics analysis. NPX values (normalized) are represented on the *y*-axis and the time point on the *x*-axis. (**B**) Venn diagram showing the overlap of differentially expressed proteins between all time points. Tables show the proteins that overlap. (**C**) Differentially expressed proteins analyzed with ELISA. (**D**) All the differentially expressed proteins from the affinity proteomics and the ELISA analysis. Lines show which proteins that have an association with hypertrophy according to the IPA’s knowledge database. Red and green colors represent up- and downregulation, respectively. Standard error of mean (SEM) is shown as error bars (*n* = 5). * = *p* < 0.05, ** = *p* < 0.01 *** = *p* < 0.001.

**Table 1 life-12-00293-t001:** ELISA kits used in the study.

ELISA Kit		
Protein	Manufacture	Cat Number
ANP	ThermoFisher Scientific	EIAANP
COL1A1	Abcam	ab210966
HSPG	Abcam	ab274393
IGFBP-7	ThermoFisher Scientific	EH252RB
PAI1	ThermoFisher Scientific	BMS2033
proBNP	ThermoFisher Scientific	EHPRONPPB

**Table 2 life-12-00293-t002:** Overlapping upregulated DEGs between 8, 24, 48, and 72 h time points with assigned gene symbol.

Gene Symbol	Entrez Gene Name
*A2M*	alpha-2-macroglobulin
*ABRA*	actin-binding Rho activating protein
*AHNAK2*	AHNAK nucleoprotein 2
*ANKRD9*	ankyrin repeat domain 9
*AQP3*	aquaporin 3 (Gill blood group)
*ARL4C*	ADP ribosylation factor-like GTPase 4C
*CALB2*	calbindin 2
*CDCP1*	CUB domain-containing protein 1
*CSDC2*	cold shock domain-containing C2
*CX3CL1*	C-X3-C motif chemokine ligand 1
*FAM83G*	family with sequence similarity 83 member G
*HK2*	hexokinase 2
*IRS2*	insulin receptor substrate 2
*NEAT1*	nuclear paraspeckle assembly transcript 1
*NEB*	nebulin
*PACSIN1*	protein kinase C and casein kinase substrate in neurons 1
*PDLIM4*	PDZ and LIM domain 4
*PDYN*	prodynorphin
*PNOC*	prepronociceptin
*PPP1R13L*	protein phosphatase 1 regulatory subunit 13 like
*PPP2R2C*	protein phosphatase 2 regulatory subunit β-gamma
*RAB11FIP4*	RAB11 family interacting protein 4
*RHOF*	ras homolog family member F, filopodia associated
*SLC12A8*	solute carrier family 12 member 8
*SLC6A17*	solute carrier family 6 member 17
*SPOCK2*	SPARC (osteonectin)-, cwcv-, and kazal-like domains proteoglycan 2
*STC1*	stanniocalcin 1
*TGFA*	transforming growth factor-alpha
*VTN*	vitronectin
*WIPF3*	WAS/WASL interacting protein family member 3

**Table 3 life-12-00293-t003:** Overlapping downregulated DEGs between 8, 24, 48, and 72 h time points with assigned gene symbol.

Gene Symbol	Entrez Gene Name
*ADRB3*	adrenoceptor beta 3
*BLM*	BLM RecQ-like helicase
*CALN1*	calneuron 1
*CCL21*	C-C motif chemokine ligand 21
*CCNE2*	cyclin E2
*CNRIP1*	cannabinoid receptor-interacting protein 1
*CTSV*	cathepsin V
*DRD1*	dopamine receptor D1
*E2F2*	E2F transcription factor 2
*FAM111B*	family with sequence similarity 111 member B
*GFRA1*	GDNF family receptor alpha 1
*HERC5*	HECT and RLD domain containing E3 ubiquitin-protein ligase 5
*HRK*	harakiri, BCL2 interacting protein
*LRRN1*	leucine-rich repeat neuronal 1
*LRRN3*	leucine-rich repeat neuronal 3
*MCM10*	minichromosomal maintenance 10 replication initiation factor
*METTL7A*	Methyltransferase-like 7A
*MTERF2*	mitochondrial transcription termination factor 2
*PDE11A*	phosphodiesterase 11A
*PRIMA1*	proline-rich membrane anchor 1
*SFRP5*	secreted frizzled-related protein 5
*SLC6A4*	solute carrier family 6 member 4
*SNCAIP*	synuclein alpha interacting protein
*STOX1*	storkhead box 1
*UNC5D*	unc-5 netrin receptor D
*XIRP2*	xin actin-binding repeat containing 2
*XRCC2*	X-ray repair cross-complementing 2

**Table 4 life-12-00293-t004:** Differentially expressed secreted proteins that were detected in the culture media. Up- and down arrows describe that the protein was significantly up- or downregulated at the given time point (FC > 1.5, *p* < 0.05).

Differentially Expressed Secreted Proteins				
Gene Symbol	8 h	24 h	48 h	72 h
*CCL4*	↑	↑		
*HBEGF*	↑	↑		
*CXCL8*	↑	↑		
*NPPB*	↑	↑	↑	
*SERPINE1*	↑	↑	↑	
*MMP1*		↓		
*CCL3*		↑		
*CXCL1*		↑		
*TNFRSF11B*		↑		
*CHI3L1*		↑	↑	
*NPPA*		↑	↑	
*CCL20*		↑		↑
*RARRES2*			↑	↑
*CCL2*			↑	
*LDLR*			↑	
*PRSS8*			↑	↑
*VEGFA*			↑	↑
*PGF*			↑	↑
*COL1A1*			↓	
*TNFRSF10B*				↑
*HMOX1*				↑
*HSPG2*				↑
*THBD*				↑

**Table 5 life-12-00293-t005:** Correlation (r-values) between mRNA and protein of ET-1 stimulated CMs. Bold text indicates a higher correlation.

Correlation between mRNA and Protein Expression after ET-1 Stimulation			
Gene Symbol	No Time-Shift	Time-Shifted	
*CCL2*	0.31	**0.95**	Increased correlation
*CCL20*	0.31	**1**	
*CCL3*	0.42	**0.99**	
*CCL4*	0.62	**1**	
*CHI3L1*	0.34	**0.77**	
*COL1A1*	0.17	**0.88**	
*CXCL8*	0.11	**0.99**	
*HMOX1*	0.03	**0.56**	
*HSPG2*	0.26	**0.88**	
*LDLR*	−0.35	**0.75**	
*MMP1*	0.43	**0.7**	
*NPPB*	0.90	**0.99**	
*PRSS8*	0.92	**0.99**	
*RARRES2*	0.72	**0.93**	
*SERPINE1*	0.83	**0.93**	
*THBD*	0.46	**0.54**	
*TNFRSF10B*	0.48	**0.87**	
*TNFRSF11B*	−0.38	**0.59**	
*CXCL1*	**0.77**	0.42	Decreased correlation
*HBEGF*	**0.95**	0.61	
*NPPA*	**0.94**	0.29	
*PGF*	**−0.13**	−0.99	
*VEGFA*	**0.75**	−0.96	

## Data Availability

Raw and processed data from RNA sequencing are available for download at ArrayExpress (https://www.ebi.ac.uk/arrayexpress/, accessed at 13 January 2022) accession number: E-MTAB-11030.

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
