# Peer review of "Multi-Omics Characterization of a Human Stem Cell-Based Model of Cardiac Hypertrophy"

_life, 2022, doi:10.3390/life12020293_

Round 1

Reviewer 1 Report

Thank you very much for the opportunity to read and review this interesting research, it’s a great pleasure.

The Article is devoted to the study of an important risk factor for the development of cardiac myopathy – cardiac hypertrophy.

During the research an in vitro model of cardiac hypertrophy based on human induced pluripotent stem cell (hiPSC) had been developed, which might be used as a relevant disease model. Approach which includes modelling processes in vitro on the basis of human pluripotent stem cells represents an attractive option that could help to uncover the mechanism underlying the development of cardiac hypertrophy on cellular level and will potentially lead to cardiovascular disease prevention and improved treatment.

Multi-omics characterization of a human stem cell-based model of cardiac hypertrophy was performed using transcriptome and secretome.

The research is elegant and well planned, and also includes numbers of replicates which makes data quite reliable. A combination of high-quality, standardized methods was applied in the research to increase the valuability of the results: flow cytometry, RNA-seq analysis, high-throughput affinity proteomics, ELISA, Ingenuity Pathway Analysis.

The obtained results are clearly and detailed presented which helps to understand complex information. Schemes of cardiac hypertrophy signaling and actin cytoskeleton signaling are highly informative, logic and quite valuable.

The following comments do not diminish the valuability of the Article.

Pages 6, 7 Figure 1 caption would be better to put on the same page with the schemes.

Line 229 Probably it would be better to show the titles of the Table 3 columns after the table break also, on the next page.

Pages 10, 11 Figure 2 caption would be better to put on the same page with the schemes.

Page 12 Probably it would be better to enlarge a bit font size of the text in a Figure 3.

Pages 12, 13 Figure 3 caption would be better to put on the same page with the schemes.

Page 15 Probably it would be better to enlarge a bit font size of the text in a Figure 5.

Line 364 Probably it would be better to change “kmeans” to “k-means”.

Page 17 Probably it would be better to enlarge a bit font size of the text in a Figure 6.

Line 505 Probably it would be better to change “βadrenergic sig-” to “β-adrenergic sig-”

It would be better to add a section “Conclusions” at the end of the Article.

It would be quite interesting if the Article would have a graphical abstract, but of course it’s on the Authors opinion.

It is necessary to formalize References according to the Journal’s requirements.

Reviewer 2 Report

Dear authors,

you put further analysis to your previous paper published in Biol Open. 2020 Sep 15; 9(9): bio052381. To the reviewers point results are clearly presented and without doubt reliable.

However, and this is a point of fine criticism to your previous paper as well, to the reviewers point of view you should describe the cultivation process more precisely, you use the cells, culture the, freeze them, reculture them and analyse. This needs to be described in more detail and arguments for the "freeze-break" should be given, as this - to the reader - is an uncommon break in the experiment. At day 19 after differentiation cells are highly positive for troponin - are theses beating cells (I would assume that "yes").

Decribe this in more detail - your model should be a "working" model (beating cells), and this make the story even more interesting, and describe the culturing in more detail, which media etc.

Reviewer 3 Report

This manuscript investigated the potential mechanisms regulating cardiac hypertrophy of via the utilization of multi-omics (transcriptomics and proteomics) analysis of cardiac hypertrophy in vitro model. 

The manuscript was well written, and the limitation of this study was well discussed as well. The analysis was clear and well described, and the results were well presented. However, the quality of figures needs to be improved.  (The resolution of Figures 3&6 is not high enough to see clearly.

One of the highlights of this study is that they linked the data to clinical pathology endpoints via IPA-Tox and identified many significantly enriched pathways. In addition, this hiPSC-CM-based model showed important characteristics of cardiac hypertrophy on transcriptional level, which could provide a feasible platform for further investigation in many other aspects. 
